# Baobab-Fruit Shell and Fibrous Filaments Are Sources of Antioxidant Dietary Fibers

**DOI:** 10.3390/molecules27175563

**Published:** 2022-08-29

**Authors:** Manuela Flavia Chiacchio, Silvia Tagliamonte, Attilio Visconti, Rosalia Ferracane, Arwa Mustafa, Paola Vitaglione

**Affiliations:** 1Department of Agricultural Sciences, University of Naples “Federico II”, 80055 Portici, Italy; 2ARWA Foodtech AB, 22363 Lund, Sweden

**Keywords:** functional foods, novel food, antioxidant capacity, procyanidins, quercetin, epicatechin, N-acylethanolamines, byproducts

## Abstract

Since 2008, baobab-fruit dried pulp is listed as an ingredient on the European Union′s Novel Food Catalogue. By pulp production, 80% of the baobab fruit is discarded, forming side streams, namely, shell, fibrous filaments, and seeds. This study explored pulp and side-stream functional properties, including total dietary fiber (TDF), total antioxidant capacity (TAC), polyphenols, and water- (WHC) and oil-holding capacities (OHC), along with endocannabinoids (ECs) and N-acylethanolamines (NAEs) in pulp, seeds, and seed oil. Shell excelled in TDF (85%), followed by fibrous filaments (79%), and showed the highest soluble and direct TAC (72 ± 0.7 and 525 ± 1.0 µmol eq. Trolox/g, respectively). Pulp was the richest in polyphenols, followed by shell, fibrous filaments, and seeds. Quercetin predominated in shell (438.7 ± 2.5 µg/g); whereas epicatechin predominated in pulp (514 ± 5.7 µg/g), fibrous filaments (197.2 ± 0.1 µg/g), and seeds (120.1 ± 0.6 µg/g); followed by procyanidin B2 that accounted for 26–40% of total polyphenols in all the products. WHC and OHC ranged between 2–7 g H_2_O-Oil/g, with fibrous filaments showing the highest values. ECs were not found, whereas NAEs were abundant in seed oil (2408.7 ± 11.1 ng/g). Baobab shell and fibrous filaments are sources of polyphenols and antioxidant dietary fibers, which support their use as functional food ingredients.

## 1. Introduction

*Adansonia digitata* L., known as the African baobab, is a wild fruit tree that grows in the arid and semi-arid lands of sub-Saharan Africa [1]. It is a socioeconomic symbol for the indigenous populations due to the extreme versatility of each part to be used as food, medicine, or object manufacturing [1].

Baobab fruit, particularly pulp, has been discovered to be an excellent source of vitamin C; minerals (as indicated in a review on composition and nutritional value of baobab food products [2]); phenolic compounds, including gallic acid, quercetin, rutin, catechin, and proanthocyanidins [3]; as well as fatty acids, such as palmitic acid, oleic acid, and linoleic acid [4]. Polyphenols and vitamin C content provide baobab pulp a soluble antioxidant capacity similar to or even higher than commonly consumed fruits such as apples, kiwis, strawberries, and oranges [5]. Moreover, its fiber content is about 70–80% of the dry mass, with pectin being the most abundant fiber [6]. Therefore, in 2008, the European Commission included the pulp in the European Union’s list of novel food ingredients, and in 2009, the Food and Drug Administration recognized it as a food ingredient [7]. Since then, its biochemical, agronomical, and botanical characteristics have been extensively studied. However, the pulp constitutes 20% of the whole baobab fruit, and the remaining 80% is made of a hard external epicarp (shell), reddish fibers covering the seeds (fibrous filaments), and the seeds [8]. Those parts represent the side streams of pulp production and are still underexplored for their techno-functional properties, which exploration of could facilitate their reutilization in the food industry as functional food ingredients. Indeed, according to the content and characteristics of bioactive molecules, agro-industrial byproducts can be used as flours, powders, or extracts to fortify other foods [9].

To the best of our knowledge, the dietary-fiber composition of baobab fibrous filaments and shell, along with the water- (WHC) and oil-holding capacities (OHC), as well as the direct total antioxidant capacity (TAC) of all the side streams, are still obscure. All the previous studies exploring the antioxidant capacity of baobab pulp and side streams focused exclusively on soluble TAC, and most studies used spectrophotometry assays to quantify the total amount of polyphenols or specific classes of compounds disregarding the characterization of phenolic compounds [10,11,12,13]. Only in one study [14] phenolic compounds of the baobab shell were characterized by liquid chromatography–mass spectrometry/quadrupole time-of-flight (LC-MS/QTOF).

Despite essential fatty acids and phosphatidylethanolamines, known precursors of endocannabinoids (ECs) and N-acylethanolamines (NAEs) [15], have been reported in baobab pulp and seeds [13], ECs and NAEs have not been investigated yet in the pulp, seeds, or seed oil. These bioactive molecules are present in food [16] and could bind receptors located in the gastrointestinal tract regulating appetite, glycaemia, nutrient metabolism, and inflammation [17].

The aim of this study was to evaluate the potential of baobab side-stream shell, fibrous filaments, and seeds in comparison with pulp, to be used in new food formulations as sources of antioxidant dietary fibers, ECs, and NAEs. Therefore, we performed a chemical characterization of the baobab products, assessing their contents in soluble dietary fiber (SDF), insoluble dietary fiber (IDF), and total dietary fiber (TDF), their direct and soluble TAC, their profile of polyphenols by high-performance liquid chromatography, and their WHC and OHC, as well as their contents of ECs and NAEs in the pulp, seeds, and seed oil.

## 2. Results

### 2.1. Total, Soluble, and Insoluble Dietary Fiber

Figure 1 shows SDF, IDF, and TDF contents of the pulp, shell, fibrous filaments, and seeds. Among the analysed samples, baobab shell was the richest in total dietary fiber (85% d.w.), followed by the fibrous filaments (79% d.w.), pulp (44% d.w.), and, finally, seeds (33% d.w.). Interestingly, pulp was mainly constituted of SDF (34% d.w.), whereas the other parts of the baobab fruit contained mainly IDF, with shell being the richest, followed by fibrous filaments.

### 2.2. Direct and Soluble Total Antioxidant Capacities

Table 1 shows the direct and soluble TAC of baobab shell, pulp, fibrous filaments, and seeds.

The soluble TAC was for shell > pulp > fibrous filaments > seeds, with the shell being 43.6-, 3.3-, and 2.1-folds higher than seeds, fibrous filaments, and pulp, respectively. Moreover, shell also showed the highest direct TAC, with a value 1.4-, 2.8-, and 11.4-folds higher than that of fibrous filaments, pulp, and seeds, respectively.

### 2.3. Characterization of Polyphenols

Table 2 shows the contents of polyphenols identified in baobab pulp, shell, fibrous filaments, and seeds.

The pulp showed the highest contents of polyphenols, followed by shell, fibrous filaments, and seeds. Epicatechin and procyanidin B2 were the most abundant compounds in pulp, fibrous filaments, and seeds, accounting for 81%, 63%, and 79% of total polyphenols, respectively; quercetin and procyanidin B2 prevailed in shell, representing together 66% of total polyphenols. Interestingly, among baobab side streams, only the shell contained procyanidin trimer in double the amount than pulp, and catechin, that was absent also in pulp. Therefore, total procyanidins accounted for 45%, 39%, 26%, and 37% of total polyphenols in pulp, shell, fibrous filaments, and seeds, respectively. Moreover, shell contained protocatechuic acid at a concentration of 21- and 3.4-folds higher than that of fibrous filaments and seeds, respectively, whereas gallic acid and rutin were at higher concentrations in fibrous filaments than in the other side streams. Quercetin-3-O-glucoside was only found in seeds.

### 2.4. Water-Holding Capacity (WHC) and Oil-Holding Capacity (OHC)

Table 3 shows the WHC and OHC in pulp, shell, fibrous filaments, and seeds. The fibrous filaments showed the highest WHC and OHC. Regarding WHC, they were followed by the shell (1.8 folds lower) and then pulp and seeds that showed similar values between each other (2.4 folds lower). Conversely, OHC was similar between pulp, shell, and seeds, being 2.7-, 2.8- and 3.6-folds lower than in fibrous filaments, respectively.

### 2.5. Endocannabinoids (Ecs) and N-Acylethanolamines (NAEs)

The endocannabinoids 2-arachidonoylglycerol (2-AG) and N-arachidonoylethanolamide (AEA) were not detected in any sample.

Table 4 shows the pulp, whole seeds, and seed oil contents of NAEs. All NAEs were at a higher concentration in the seed oil than in the whole seeds (with the exception of OEA), with the pulp showing the lowest. Linoleylethanolamide (LEA) was the most abundant compound in all the samples, especially in seed oil. Stearoylethanolamide (SEA) was the least abundant, both in pulp and whole seeds, whereas Oleoylethanolamide (OEA) was the least abundant within the seed oil. Total NAEs concentration in seed oil was 3-folds higher than seeds, which, in turn, was 3-folds more concentrated than pulp.

## 3. Discussions

Nutritional valorization of agro-industry byproducts and their reutilization in the food chain is a fruitful strategy to reduce food waste and enhance food-chain sustainability.

For the first time, this study showed that baobab fibrous filaments and shell are rich sources of dietary fiber (79–85%), mostly insoluble, and have a relevant antioxidant capacity (both soluble and direct) associated with the polyphenol content. Pulp contained 44% of fiber, mainly soluble, whereas seeds had 33% of fiber, which agrees with previous works [18,19]. The relative amount of soluble and insoluble dietary fiber may affect technological properties of the fibers when used in food formulation, along with the physiological benefits upon their consumption [20]. High soluble dietary fibers often possess also a high WHC [20]. Therefore, fibrous filaments containing 8.6% of SDF showed a higher WHC than in shell (2.3% SDF). This feature makes fibrous filaments useful to control syneresis and modify the viscosity and consistency of food formulations [21] and to prolong the chewing time of foods, increase fecal volume, and affect appetite cues [22]. Conversely, highly insoluble dietary fibers, which mainly show a high OHC [20], are useful ingredients for stabilizing food emulsions and for decreasing digestion and lipid absorption along the gastrointestinal tract [21]. Indeed, an ingredient that binds oil/fat in the food matrix can prevent lipid loss during food processing and cooking, thus improving the texture, sensory aspects, and shelf life of food with a high percentage of fat and emulsion; while, if lipid binding to the ingredient occurs during the digestion (after food consumption), it may reduce blood lipid levels, affecting their intestinal absorption [21,23].

Interestingly, the WHC and OHC of baobab fruit pulp and side streams were similar to oat bran, wheat bran, pea cortical, pomegranate peel [24], and coffee byproducts [25], whereas the OHC of baobab fibrous filaments, pulp, shell, and seeds was, on average, from 3- to 0.9-folds higher than potato fiber, pea cortical, carrot fiber, barley fiber, and pomegranate peel [24].

The high content of IDF, along with the functional capacity to bind water and oil, support the reutilization of baobab side streams in food formulations, possibly advantaging both technological aspects and human health upon consumption.

The direct TAC shown by the baobab shell and fibrous filaments, along with the contents of IDF and polyphenols, indicated that they are antioxidant-dietary-fiber-rich products, i.e., they contain natural antioxidants, like polyphenols, associated with the fiber matrix [26].

Several plant-based foods, such as vegetables, fruits, cereals, and seeds (like cocoa bean and coffee bean), and agricultural byproducts provide antioxidant dietary fiber [26,27]. Interestingly, baobab side streams (from food production line) and pulp showed a direct TAC that was, on average, from 0.1- to 1.2- folds higher than common foods, such as whole wheat, apple, spinach, bean, and seeds from pomegranate fruit, analyzed in previous studies [28,29]. This observation may be functionally relevant upon consumption because antioxidant dietary fiber may exert antioxidant activity along the gastrointestinal tract, enter the colon, and interact with the gut microbiota [27,30,31].

Fibrous filaments and seeds mainly contain epicatechin and procyanidin B2, similarly to pulp, which agreed with a previous study [3]. Compared to pulp and the other side streams, the shell contains higher concentrations of procyanidin trimer, protocatechuic acid, and quercetin, which explains the higher TAC of this product compared to the pulp and the other side streams. The radical scavenging capacity is increased in polymerized procyanidins compared to their constituent monomers (catechin and epicatechin) and in flavonoids and phenolic acids containing a catechol group on the B-ring structure and benzene, as in quercetin and protocatechuic acid, respectively; conversely, glycosylation of flavonoids reduces greatly the radical scavenging capacity, as in rutin compared to quercetin [32]. Procyanidins are largely distributed in agro-industrial byproducts such as coffee and cocoa hulls, as well as grape seeds, supporting their antioxidant, antibiotic, and anti-inflammatory properties [33], whereas quercetin in citrus peel supported antihyperglycemic and antihyperlipidemic effects of this product [34].

The polyphenol content in baobab side streams could justify their utilization as ingredients in the food formulation directly or after extraction. Previous studies showed the ability of some byproducts, i.e., green walnut hulls and cocoa bean shell, to stabilize color and reduce oxidation in meat products or to increase antioxidant content in bakery products [35,36]. Moreover, polyphenol extract from fruit byproducts such as mango seed kernels were used to develop active packaging with antimicrobial and antioxidant properties [37].

In the present study, for the first time, ECs and NAEs contents were assessed in baobab pulp, whole seeds, and in the seed oil. The absence of endocannabinoids 2-AG and AEA was expected, as a previous study found that they are mainly (but not exclusively) present in animal products [16]. The higher content of NAEs in seed oil than seeds and pulp was also expected due to the lipid nature of these molecules. Blood levels of ECs and NAEs are metabolically linked to the dietary fat and are tightly involved in energy homeostasis regulation, appetite stimulation, pain sensation, inflammation, immunity, health, and well-being [17,38]. Some evidence suggests that dietary ECs and NAEs may bind receptors located in the gastrointestinal tract [16,38] and regulate appetite, nutrient metabolism, and inflammation from the intestinal lumen; however, the physiological relevance of dietary ECs and NAEs is still underexplored.

The fatty acid composition and antioxidant contents of baobab seed oil are particularly appreciated [39]. Our findings reveal that baobab seed oil contains a higher amount of NAEs than the extra-virgin olive oil, hemp oil, and coconut oil previously analyzed by [16]. This observation supports future studies aiming to evaluate the effects on health from consuming baobab seed oil and the contribution eventually provided by NAEs content in modulating appetite and satiety [17]. Before that, to allow human consumption, baobab seed oil must be opportunely treated to reduce or remove cyclopropenoid fatty acids, such as sterculic acid and malvalic acid, which are toxic compounds that render the oil inedible as it is, according to the World Health Organization (WHO) and Food and Agriculture Organization (FAO) [40]. Therefore, future studies are needed before considering the possibility to use baobab seed oil as a functional oil in new food formulations.

## 4. Materials and Methods

### 4.1. Chemical

The reagents used were ABTS (2,2-Azinobis (3-ethylbenzothiazoline-6-sulfonic acid) (Sigma-Aldrich, Milan, Italy) (A1888), potassium persulfate (Sigma-Aldrich, Milan, Italy) (379824), ethanol (Sigma-Aldrich, Milan, Italy) (1.59010), methanol (Sigma-Aldrich, Milan, Italy) (34860), water (Sigma-Aldrich, Milan, Italy) (270733), formic acid (Sigma-Aldrich, Milan, Italy) (F0507), acetonitrile (Sigma-Aldrich, Milan, Italy) (34851), chloroform (Sigma-Aldrich, Milan, Italy) (C2432), HCl 12 M (Sigma-Aldrich, Milan, Italy) (320331), Trolox (Sigma-Aldrich, Milan, Italy) (238813), gallic acid (Sigma-Aldrich, Milan, Italy) (91215), chlorogenic acid (Sigma-Aldrich, Milan, Italy) (C3878), ellagic acid (Sigma-Aldrich, Milan, Italy) (2250), epicatechin (Sigma-Aldrich, Milan, Italy) (E1753), catechin (Sigma-Aldrich, Milan, Italy) (43412), quercetin (Sigma-Aldrich, Milan, Italy) (Q4951), caffeic acid (Sigma-Aldrich, Milan, Italy) (C0625), rutin (Sigma-Aldrich, Milan, Italy) (R5143), sodium chloride (Sigma-Aldrich, Milan, Italy) (S9888), sodium anhydrous sulphate (Sigma-Aldrich, Milan, Italy) (PHR2658), tris (Sigma-Aldrich, Milan, Italy) (T1503), mes (Sigma-Aldrich, Milan, Italy) (M3671), acetone (Sigma-Aldrich, Milan, Italy) (179124), and NaOH 6M (Sigma-Aldrich, Milan, Italy) (567530). A total-dietary-fiber assay kit including α-amylase, protease, and α-amyloglucosidase for the starch and protein hydrolysis was purchased from Megazyme (Wicklow, Ireland) (K-TDFR-200A). Cellulose powder from spruce was obtained from Fluka (Steinheim, Germany). The ECs (Arachidonoylethanolamide d8; AEA d8; 2-Arachidonoylglycerol, 2-AG; arachidonoylethanolamide, AEA) were purchased from Cayman Chemical (Ann Arbor, MI, USA) (390050,62160, 90050). The NAEs (oleoylethanolamide, OEA; linoleoylethanolamide, LEA; palmitoylethanolamide, PEA) standards were purchased from Cayman Chemical (Ann Arbor, MI, USA) (90265, 90155, 90350). Sunflower seed oil was purchased from a local market.

### 4.2. Baobab Samples

The parts of baobab fruit used in this study were pulp, seeds, fibrous filaments, and shell (Figure 2). They were obtained from one batch of fruits and were supplied by ARWA Foodtech AB (Lund, Sweden).

The seeds were frozen, freeze-dried by a Heto LyoLab 3000 (Thermo scientific Heto, Denmark), and then ground using a GRINDOMIX 2000 (Retsch Italia, Verdere Scientific S.r.l, Bergamo, Italy). Ground seeds were defatted with n-hexane (20 mL per g seeds), and defatted seeds were used for dietary-fiber analysis. Seeds were also extracted to obtain oil [41]. Each sample (5 g) was weighed in triplicate and placed in a Falcon tube with 20 mL of a chloroform:methanol (2:1) solution. The mixture was shaken on a tilting laboratory shaker for 20 min and then centrifuged for 10 min. The supernatant was removed and placed into a separating funnel; the extraction was repeated four times. Thereafter, 40 mL of a 0.5% sodium chloride solution was added to the separating funnel, and a vigorous shaking for 1 min was applied. After 5 min, the chloroform phase was collected in a Falcon tube, 15 g of sodium anhydrous sulfate was added, and the mixture was filtered through a filter paper. The lipid extract was obtained by evaporation of the chloroform using a Rotavapor. From 5 g of dry sample, we obtained 0.79 g of oil with a total extraction yield of 16%.

### 4.3. Soluble, Insoluble, and Total Dietary Fiber

The SDF and IDF contents of all the samples were determined according to a gravimetric–enzymatic method [42] with brief modifications. Samples (0.5 g) underwent enzymatic digestion to hydrolyze starch and proteins. Then, the mixture was filtered through a crucible, and the retentate was washed with 78% ethanol, pure ethanol, and acetone to obtain the IDF. The permeate was added to ethanol and maintained at 60 °C for 1 h to promote the precipitation of the SDF that was separated by filtration and washed as described above. The crucibles with retentates were dried overnight at 105 °C in an air oven and weighed to calculate the SDF and IDF. Final values were corrected for ashes and proteins in the sample. The analysis of each sample was performed in triplicate. Total dietary fiber, SDF, and IDF were expressed as percentages of the whole sample weight.

### 4.4. Direct and Soluble Total Antioxidant Capacities

The direct TAC was measured by the ABTS–QUENCHER method [43]. When necessary, dilution was performed by mixing directly the solid sieved sample with cellulose, depending on the antioxidant capacity of the samples. The reaction was started by adding 6 mL of ABTS^•+^ to 10 mg of the diluted sample. After 30 min, samples were centrifuged, and the absorbance values were measured at 734 nm. The same procedure was carried out for the blank using 10 mg of cellulose. Each sample was analyzed in triplicate.

The soluble TAC was assessed by the ABTS^•+^ method [44] using extracts obtained from 1 g of each sample with 6 mL of 0.1% acidified MeOH, vortexing for 1 min, sonicating for 30 min, and centrifuging for 10 min at 4000 rpm and 4 °C. The supernatants were collected in dark tubes and used for the ABTS^•+^ assay. The reaction was started by adding 1 mL of ABTS^•+^ to 100 µL of the extract. After 2.5 min, the absorbance was measured at 734 nm. The same procedure was carried out for the blank using 100 µL of methanol instead of the extract. Three extracts for each sample were prepared and analyzed.

The results for direct and soluble TAC were expressed as µmol eq. Trolox/g of dry sample using a Trolox calibration curve.

### 4.5. Polyphenol Characterization by HPLC and HPLC-MS/MS Analysis

The determination of polyphenols in baobab pulp, shell, fibrous filaments, and seeds was carried out following the method proposed by [24] with slight modifications. One gram of each sample was weighed, and 10 mL of 0.1% acidified MeOH/water solution (70:30 *v*/*v*) was added. After vortexing for 1 min and sonicating for 30 min, the mixture was centrifuged for 10 min at 14,800 rpm and 4 °C, and the supernatant was collected in dark tubes and filtered with 0.45 µm PTFE filters before the HPLC analysis. Three extracts of each sample were prepared and analyzed.

The chromatographic separation was performed according to [45] using a HPLC SHIMADZU with UV/VIS SPD-20° (Prominence, CA, USA) as a detector set at 280 nm. The mobile phases were a 0.2% formic acid water (solvent A) and acetonitrile/methanol (60:40 *v*/*v*) (solvent B). A Prodigy ODS3 100 Å (250 mm × 4.6 mm, particle size 5 μm) column (Phenomenex, CA, USA) was used. An injection volume of 20 µL was used for each run with a constant flow of 1 mL/min. The gradient program was set as follows: 20% B (0–2 min), 20–30% B (6 min), 30–40% B (10 min); 40–50% of B (8 min), 50–90% of B (8 min), constant flow to 90% of B (3 min); and, to rebalance the column, 90–20% of B for 2 min and 20–20% of B for 4 min. Each sample was injected three times to obtain three independent replicates. Gallic acid, chlorogenic acid, epicatechin, caffeic acid, rutin, and quercetin were identified by the corresponding standard compounds.

To confirm the identification and to identify the peaks remained unknown after HPLC analysis, a HPLC-MS/MS analysis of the extracts was also performed. An API 3000 Triple Quadrupole mass spectrometer (Applied Biosystem Sciex, Foster City, CA, USA) equipped with a TurboIonSpray source was used. Analysis was performed in negative-ion mode and in MRM (multiple-reaction mode). Mass-spectrometry conditions were the same set as for the HPLC analysis and according to [45]. Protocatechuic acid, procyanidin B2, procyanidin trimer, and quercetin 3-*O*-glucoside were identified by HPLC-MS/MS analysis. Table 5 shows the retention times of polyphenols identified in the samples, whereas HPLC-MS/MS acquisition parameters of monitored compounds are reported in Table 6. The extracted-ion chromatograms (XIC) of baobab pulp, shell, fibrous filaments, and seeds confirmed by HPLC-MS/MS are shown in Appendix A.

The quantification of all the identified compounds was performed by HPLC analysis. Specifically, gallic acid, chlorogenic acid, catechin, epicatechin, caffeic acid, rutin, and quercetin were quantified using the calibration curves of the corresponding standard compounds; protocatechuic acid was quantified as a caffeic acid equivalent, procyanidin B2 and procyanidin trimer as catechin equivalent, and quercetin 3-*O*-glucoside as quercetin equivalent.

### 4.6. Water- and Oil-Holding Capacities

The water-holding capacity (WHC) and oil-holding capacity (OHC) of baobab pulp, shell, and seeds were determined as described by [24]. Each sample (200 mg) was mixed with 12 mL of distilled water for the determination of WHC and with 12 mL of sunflower oil for the OHC measure. After 24 h of stirring at room temperature, samples were centrifuged at 4000 rpm for 45 min. Then, the supernatant was carefully removed, and the wet pellet was weighed. WHC and OHC were calculated as grams of water/oil retained per gram of the sample. Three replicates were made for each sample.

### 4.7. Determination of Endocannabinoids (ECs) and N-Acylethanolamines (NAEs)

Extractions of ECs and NAEs from pulp, seeds, and seed oil were performed by the method described by Bligh and Dyer [41], with brief modifications. Samples (100 mg) were added to 50 μL of the internal standard 2 µg/mL solution of Arachidonoylethanolamide d8 (AEA d8). A volume of 1.5 mL of chloroform/methanol (2:1) was added to the samples, which were vortexed for 20 s and centrifuged at 14,800 rpm for 10 min at 4 °C. The supernatants were then collected in a glass tube, and the pellets were extracted with a chloroform/methanol (2:1) solution twice more. The chloroform was evaporated under nitrogen and reconstituted in a 200 μL acetonitrile/isopropanol/water (60:35:5) solution prior to high-performance liquid chromatography/high-resolution mass spectrometry (HPLC-HRMS) analysis.

HPLC-HRMS analysis of extracts was performed according to [38]. Data were collected using an Accela U-HPLC System, consisting of a quaternary pump and a thermostated autosampler (10 °C) coupled to an Exactive Orbitrap MS equipped with a heated electrospray interface (HESI) (Thermo Fisher Scientific, San Jose, CA, USA). Compounds were separated on a Kinetex 2.6 μ C18 100 Å (100 mm × 2.1 mm) column (Phenomenex, Torrance, CA, USA) with a set temperature of 45 °C and eluted by a linear gradient of a 40:60 water/acetonitrile solution (5 mM ammonium formate and 0.1% formic acid) (solvent A) and 90:10 isopropanol/acetonitrile (5 mM ammonium formate and 0.1% formic acid) (solvent B) with a rate of 200 μL/min and injection volume of 10 μL. The elution gradient was set according to [29]. Detection was performed in the positive range. The compounds were identified and quantified according to authentic standards using the exact mass value to the decimal place (mass tolerance ± 5 ppm). Standards for ECs (2-AG; AEA; AEAd8) and NAEs (OEA, LEA, PEA) were purchased from Cayman (Cayman Chemical, Ann Arbor, MI, USA).

### 4.8. Data Analysis

All the analysis were performed in three replicates, and the results were expressed as mean ± standard deviation. Statistical analysis was performed using statistical software SPSS (version 20.0, SPSS, Inc., Chicago, IL, USA). The differences between samples were assessed by the one-way ANOVA and Tukey’s post hoc tests. Two-tailed *p*-values lower than 0.05 were considered significantly different. Data were expressed as means ± SD.

## 5. Conclusions

In this study, some functional properties of baobab pulp, shell, fibrous filaments, seeds, and seed oil were investigated. Baobab shell and fibrous filaments excelled in fiber content (mainly IDF) and antioxidant capacity, both soluble and direct (i.e., exerted by the whole product powders, prior to extraction). This observation indicated that the two baobab side streams provide a greater amount of antioxidant dietary fiber than pulp and revealed opportunities for their use as functional food ingredients to counteract oxidative stress along the gastrointestinal tract and to modulate the gut microbiota.

Among baobab side streams, the shell was the richest in polyphenols and showed the highest soluble antioxidant capacity, possibly due to the content in procyanidin trimer, procyanidin B2, protocatechuic acid, and quercetin. Fibrous filaments were distinguished for their capacity to bind water and oil, thus indicating they may find application in food to control syneresis and modify the viscosity and consistency, as well as stabilize food rich in fat and emulsion. Seed oil displayed an amount of NAEs higher than whole seeds and pulp, thus possibly having anorexigenic effect upon consumption; however, further studies are needed on this aspect.

Altogether, findings of this study indicate that shell and fibrous filaments are antioxidant-dietary-fiber-rich ingredients showing functional properties that support their direct reutilization in the food chain as functional food ingredients that may provide valuable nutritional and technological advantages. Specifically designed studies are needed to demonstrate the effectiveness of those side streams in real food systems.

## Figures and Tables

**Figure 1 molecules-27-05563-f001:**
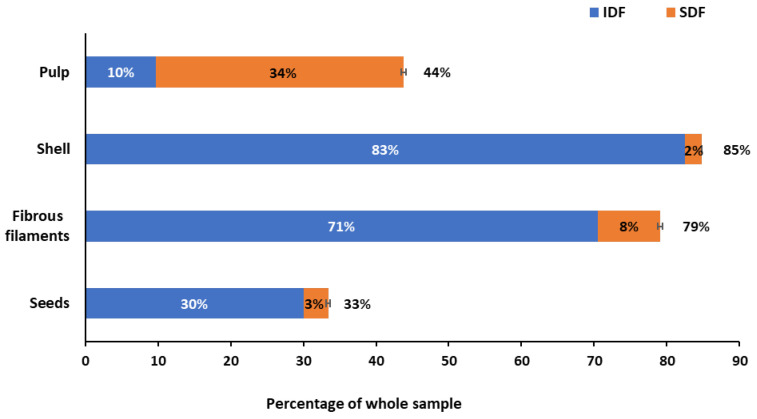
Insoluble (IDF, blue bar), soluble (SDF, orange bar), and total dietary-fiber (TDF, whole bar) contents in baobab pulp, shell, fibrous filaments, and seeds. Data are expressed as % d.w. of the whole sample (mean ± SD). The numbers on blue bars indicate the IDF, on orange bars the SDF, and on the top of the whole bars the TDF.

**Figure 2 molecules-27-05563-f002:**
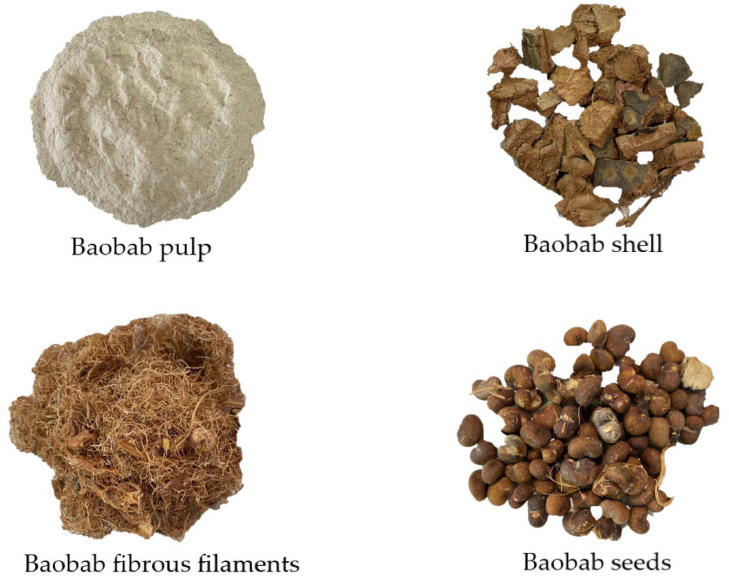
Baobab pulp, baobab shell, baobab fibrous filaments, and baobab seeds.

**Table 1 molecules-27-05563-t001:** Direct and soluble total antioxidant capacities (TAC). Data are shown as µmol eq. Trolox/g d.w. for TAC (mean ± SD). Different letters indicate differences between samples in the same column (*p* < 0.05).

Samples	Direct TAC	Soluble TAC
Pulp	187.6 ± 2.0 ^c^	37.4 ± 0.2 ^b^
Shell	524.9 ± 1.0 ^a^	72.0 ± 0.7 ^a^
Fibrous filaments	366.3 ± 8.6 ^b^	21.9 ± 0.9 ^c^
Seeds	45.8 ± 0.0 ^d^	1.6 ± 0.1 ^d^

**Table 2 molecules-27-05563-t002:** Polyphenols in baobab pulp, shell, fibrous filaments, and seeds. Data are shown as µg/g d.w. (mean ± SD). Different letters indicate differences between samples in the same row (*p* < 0.05).

Polyphenols	Pulp	Shell	Fibrous Filaments	Seeds
Gallic acid	33.9 ± 1.1 ^c^	78.3 ± 0.3 ^b^	88.5 ± 0.4 ^a^	19.6 ± 0.8 ^d^
Protocatechuic acid	-	50.0 ± 0.2 ^a^	2.4 ± 0.005 ^c^	14.7 ± 1.3 ^b^
Chlorogenic acid	1.8 ± 0.09 ^a^	-	-	-
Catechin	-	47 ± 0.1 ^a^	-	-
Epicatechin	514 ± 1 ^a^	81.9 ± 0.7 ^d^	197.2 ± 0.1 ^b^	120.1 ± 0.6 ^c^
Procyanidin B2	506.5 ± 5.7 ^a^	336.7 ± 1.0 ^b^	135.5 ± 0.4 ^c^	106.3 ± 0.02 ^d^
Rutin	80.3 ± 0.9 ^a^	17.5 ± 0.5 ^c^	70.3 ± 0.1 ^b^	1.5 ± 0.03 ^d^
Caffeic acid	25.9 ± 0.8 ^a^	-	6 ± 0.006 ^b^	3.9 ± 0.01 ^c^
Procyanidin trimer	60 ± 1.4 ^b^	112.3 ± 1.5 ^a^	-	-
Quercetin 3-*O*-glucoside	-	-	-	15.5 ± 0.7 ^a^
Quercetin	39.5 ± 1.4 ^b^	438.7 ± 2.5 ^a^	28.5 ± 0.03 ^c^	4.6 ± 0.004 ^d^
TOTAL	1262 ± 5.8 ^a^	1162 ± 0.5 ^b^	528.5 ± 1 ^c^	286.3 ± 0.8 ^d^

**Table 3 molecules-27-05563-t003:** Water-holding capacity (WHC) and oil-holding capacity (OHC) in baobab samples. Data are shown as g(H_2_O-Oil)/g d.w. (mean ± SD). Different letters indicate differences between samples in the same column (*p* < 0.05).

Samples	WHC	OHC
Pulp	3.2 ± 0.2 ^c^	2.7 ± 0.0 ^b^
Shell	4.3 ± 0.1 ^b^	2.5 ± 0.1 ^b^
Fibrous filaments	7.7 ± 0.2 ^a^	7.2± 0.5 ^a^
Seeds	3.2 ± 0.3 ^c^	2.0 ± 0.2 ^b^

**Table 4 molecules-27-05563-t004:** Linoylethanolamide (LEA), palmitoylethanolamide (PEA), oleoylethanolamide (OEA), stearoylethanolamide (SEA), and total N-acylethanolamines (NAEs) concentrations in baobab samples. Data are shown as ng NAEs/g d.w. (mean ± SD). Different lowercase letters indicate significant differences between NAEs within the sample (in the same row) (*p* < 0.05). Different uppercase letters indicate differences between samples (in the same column) (*p* < 0.05).

Samples	LEA	PEA	OEA	SEA
Pulp	107.9 ± 0.3 ^C,a^	46.3 ± 1.1 ^C,b^	44.7 ± 1.5 ^C,b^	36.9 ± 0.1 ^C,c^
Seeds	325.2 ± 14.1 ^B,a^	174.3 ± 13.1 ^B,c^	218.7 ± 0.1 ^A,b^	67.8 ± 3.1 ^B,d^
Seed oil	995.3 ± 21.1 ^A,a^	520.9 ± 5.5 ^A,c^	152.9 ± 1.6 ^B,d^	739.5 ± 2.8 ^A,b^

**Table 5 molecules-27-05563-t005:** Retention time of polyphenols identified in the samples by HPLC analysis.

Polyphenols	Retention Time (min)
Gallic acid	3.8
Protocatechuic acid	5.7
Chlorogenic acid	6.8
Procyanidin trimer	6.5
Catechin	7.1
Procyanidin B2	7.3
Epicatechin	9.7
Caffeic acid	10.2
Quercetin 3-*O*-glucoside	10.6
Rutin	14.5
Quercetin	26.6

**Table 6 molecules-27-05563-t006:** HPLC-MS/MS acquisition parameters of monitored compounds.

Compound	Precursor Ions [M − H]^-^ (*m*/*z*)	Product Ions (*m*/*z*)
Gallic acid	169	125, 79
Protocatechuic acid	153	109
Chlorogenic acid	353	191
Procyanidin trimer	865	577, 451, 695, 289
Catechin and Epicatechin	289	245
Procyanidin B2	577	125, 425, 407, 289
Caffeic acid	179	135
Quercetin 3-*O*-glucoside	463	301, 300, 271
Rutin	609	301, 271
Quercetin	301	151, 179

## Data Availability

The data presented in this study are available on request from the corresponding author.

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
