# Peer review of "Baobab-Fruit Shell and Fibrous Filaments Are Sources of Antioxidant Dietary Fibers"

_molecules, 2022, doi:10.3390/molecules27175563_

Round 1

Reviewer 1 Report

The introduction is well written.

Results

The results for dietary fibres are a bit confusing. You must present results as percentages of total, insoluble and soluble dietary fibres per whole sample.

Table 3 - Correct H20, make 2 into an index.

In the whole manuscript, the p-value needs to be in italic.

Discussion

Line 159 - what are the common foods? It is unclear.

Materials and methods

Determination of dietary fibres: Was the content of fibres determined in the seeds before or after oil extraction? Because this method is affected by high contents of fat.

In how many repetitions were all methods performed? There is only information for OBC and WBC.

Conclusion

The conclusion is too general. What does it mean "showed interesting properties when compared to other commonly consumed food matrices"? You need to explain more what are the main conclusions of this study.

Reviewer 2 Report

In this study, authors measured some technofunctional properties and bioactive components in pulp and byproducts from Adansonia digitata L. (Baobab fruit). Mainly phenolic compounds, fiber, endocannabinoids (ECs) and N-acylethanolamines (NAEs), as well as water and oil holding capacity. However, besides the partial characterization of fibrous filaments, I am not sure if the scientific relevance and novelty is enough for publication in this journal. Numerous papers assessing the metabolic profile, functional potential and other applications of shells have been published, indeed, authors must highlight novelty and impact of their article. Full characterization would be more interesting and novel than just measuring some compounds and functionality. Since pulp is a food ingredient, I consider mandatory a proximate characterization. 

Please, indicate how this article differentiates from the following (highlight novelty and impact): https://www.scirp.org/journal/paperinformation.aspx?paperid=110646

Discussion is speculative in some parts, authors should provide evidence to support some hypothesis they have stated.

Introduction

Authors should include more information regarding endocannabinoids and N-acylethanolamines. Why these specific compounds are of interest in baobab fruit? Include the state-of-the-art, potential uses and perspectives.

By shell you mean peel?

Please describe what you mean by “fibrous filaments”. Is it the same as bagasse?

Results

Besides the fiber content, the most interesting part of this work is the MS analysis. However, characterization of metabolites in shells (limited to 10 compounds) does not provide novel findings, with some constituents being solely tentatively identified. Other works reported 45 compounds for baobab shells (https://doi.org/10.1016/j.scitotenv.2019.07.193), at least 13 in pulp https://doi.org/10.1016/j.foodres.2017.06.025, https://doi.org/10.1016/j.foodchem.2016.07.005,

Pictures of the fruit and its by-products would be very illustrative.

Please provide the chromatograms of the HPLC-ESI-TOF/MS in the main manuscript or supplement file.

If LC-MS/MS was carried out, why authors did not explore other identified compounds and not only those used for quantitation?

Please change LC-MS/MS by HPLC-MS/MS throughout the manuscript

L146-150 Which are the potencial implications or applications of high OHC? Please describe

L152 - Authors cannot claim that baobab is source of antioxidant fiber. Please review the definition of antioxidant fiber and the requirements to be considered a “good source”

L153 - Authors declare that the fruit has “significant amounts” of antioxidants, including polyphenols. Please clarify what is the point of reference for claiming this.

L181 - Authors need to clarify the importance of quantifying ECs and NAEs

Table 6 does not show conditions of analysis, it shows the identified compounds by LC-MS. Please include complete information, for example: accurate mass, λmax (nm), wavelength used for detection of each compound.

L182-183 If 2-AG is present only in animal samples, why did the authors measure this compound?

L195- Since cyclopropenoid fatty acids are hazardous for human health, I consider their quantification is essential.

L198 - Please provide a potencial use for the oil

Fig 1. Bar plot should be more illustrative

Methods

L208 - Provide full information of the product, cat number at least

L214-217 - Did the samples belong to a single batch? Or different batches were taken overtime to be more representative? how many replicates were analyzed?

L254 - Section 4.5.1 should be placed before section 4.4. Does this method allow the efficient recovery of total polyphenols? This information might be useful: https://doi.org/10.1016/j.sajb.2019.01.034, https://doi.org/10.1016/j.scitotenv.2019.07.193

Conclusion

This section must be improved, it is not adequate just summarize the study.

L333 - Please review this claim

Reviewer 3 Report

The paper titled "Baobab fruit shell and fibrous filaments are valuable sources of antioxidant dietary fibers" investigated pulp and side stream functional properties including dietary fiber, total antioxidant capacity, polyphenols, water- and oil-holding capacity along with endocannabinoids (ECs) and N-acylethanolamines (NAEs) in pulp, seeds and seed oil. The results of the present study are interesting and help for enhancing nutritional and technological advantages of Baobab side stream as functional ingredients in food industry. However, some issues should be considered.

In the Introduction part, I would recommend the authors to give some background information about endocannabinoids (ECs) and N-acylethanolamines (NAEs). What are their important functions?

Line 65, for more intuitive expression, Figure 1 should be drawn as a stack column.

Line 98, please check and revise the sentence into "...by the shell (1.8 folds lower)...".

Line 99, please check and revise the sentence into "...between each other (2.4 folds lower)...".

Lines 109-110, OEA concentration in the seed oil is not higher than in the whole seeds. Please revise the sentence.

Line 113, OEA was the least abundant in pulp. Please revise the sentence.

Line 261, LC-MS/MS analysis? Please check this subtitle.

Line 305, please give the full name of LC-HRMS.

Round 2

Reviewer 1 Report

Thank you to the authors for accepting my suggestions for this manuscript.

Reviewer 2 Report

Authors addressed all the observations. I consider it can be accepted in its current form.